# Psychological Implicit Motives Construct as an Emergent Fractal Attractor from Intermittent Neurophysiological Responses: Simulation and Entropy-like Characterization

**DOI:** 10.3390/e25050711

**Published:** 2023-04-25

**Authors:** Miguel Ángel Martín, Celia Vara, Carlos García-Gutiérrez

**Affiliations:** 1Department of Applied Mathematics, Universidad Politécnica de Madrid, 28040 Madrid, Spain; 2Department of English, McGill University, Montreal, QC H3A 0G4, Canada

**Keywords:** Implicit Motives, Iterated Random Function Systems, information entropy

## Abstract

Implicit Motives are non-conscious needs that drive human behavior towards the achievement of incentives that are affectively incited. Repeated affective experiences providing satisfying rewards have been held responsible for the building of Implicit Motives. Responses to rewarding experiences have a biological basis via close connections with neurophysiological systems controlling neurohormone release. We propose an iteration random function system acting in a metric space to model experience–reward interactions. This model is based on key facts of Implicit Motive theory reported in a broad number of studies. The model shows how (random) responses produced by intermittent random experiences create a well-defined probability distribution on an attractor, thus providing an insight into the underlying mechanism leading to the emergence of Implicit Motives as psychological structures. Implicit Motives’ robustness and resilience properties appear theoretically explained by the model. The model also provides uncertainty entropy-like parameters to characterize Implicit Motives which hopefully might be useful, beyond the mere theoretical frame, when used in combination with neurophysiological methods.

## 1. Introduction

The Complex System approach [1] has been extremely useful to understand a broad range of natural structures and phenomena in different areas of science. One way of recognizing the complex nature of a given entity might consist of understanding the underlying mechanism which generated it and to study whether such a mechanism fits some mathematical scheme that explains the roots of complexity. This might allow the application of different concepts, tools, and theoretical results from the Mathematics of Complexity and hopefully find applications of interest in the field where the entity is located.

This work attempts to address the mathematical modeling of the Implicit Motive (IM) construct within the well-established Motive Disposition Theory (MDT) [2,3]. IMs are considered a central part of the psychological motivational system and have a great influence on human behavior [4]. These motives are assumed to operate outside of conscious awareness and control, and implicit measures are adopted [5].

There are three major and fundamental needs that energize and direct behavior within the Implicit Motives system. These are the need for Achievement (*n* Ach), the need for Power (*n* Pow), and the need for Affiliation (*n* Aff). The first one is the need to reach a standard of excellence while avoiding goals that are excessively difficult or too easily achieved [6]. The need for Power is the motivation to exert influence over others. The use of power may be manipulative and controlling but may also be used to help and support others [7]. The Affiliation motive is the desire to establish and maintain close bonds and a sense of belonging through individual relationships and social connectedness [8]. These basic needs are considered fundamental to all humans but are assumed to differ in strength between cultures and individuals [2], and they represent a capacity to derive satisfaction from the attainment of the above-mentioned domain-specific incentives (e.g., mastering a challenge) and avoidance of certain classes of disincentives [4]. Comprehensive reviews of research have explored the validity of the IM construct [4,9,10,11,12,13].

A person’s implicit motivational disposition is referred to as their Implicit Motives. Adults can have very different IMs that may shape their behavior and career [4], but it is agreed that these dispositions develop early in the childhood stage. Some studies relate each need’s strength in adulthood to childhood rearing practices. For example, individuals with high nAch have been observed to be raised by parents who set age-appropriate challenging tasks and reward the children’s autonomous task mastery with affection [14]. A longitudinal study [15] indicates that there could be some relation between high nAff adult’s scores to the mother’s unresponsiveness to the child’s crying, but this relation is not conclusive. Finally, nPow can be affected by both prenatal hormone exposure and parental permissiveness for aggressive and sexual behavior [4].

In summary, researchers postulate that IMs are built and fine-tuned as a result of repeated affective experiences yielding satisfying rewards in early childhood [4,5,9,15]. In particular, the hormonal cascade produced by the succession of repeated rewarding experiences should be at the core of how IMs are built [2,9]. This does not come as a surprise, as Implicit Motives show close connections with physiological systems controlled by midbrain structures such as neurohormone release [2,4,16,17,18,19,20,21]. For a review, see [22].

Once IMs are developed, there are several ways to assess the predominance of the needs in an individual, e.g., the Picture Story Exercise [23] or the Thematic Apperception Test [24]. However, these methodologies are not without criticism [5,15] and the objective of this paper goes in another direction.

Underlying the validity of the IM psychological identity, there is an implicit recognition of structural organization, which is consistent with considering the brain as a self-organizing system [25,26]. Under the point of view of self-organization theories, the unsupervised learning through early intermittent rewarding experiences and their modular storage would create a global (unconscious) stable structure [27]. In this way, the IM construct has been solidly established within the psychology field and “has become a poster child for the natural-science type of validity” [4,28]. The present work aims to be a contribution in the direction of supporting the robustness of the construct from the mathematical point of view.

This paper presents a mathematical model for the development of IM whose aim is two-fold. First, it can provide insight into the underlying mechanism leading to the emergence of these psychological structures and on its stability properties; second, it provides entropy-like uncertainty parameters that can continuously characterize IMs.

## 2. Methods

A simple mathematical model is formulated to account for the rewarding dynamics leading to Implicit Motives strictly based on key theses within the Motive Disposition Theory. These are:

**Thesis** **1.**Repeated responses to affective-emotional experiences are the responsible iterated actions on which IMs are built. Any new experience would be supposed to leave a physiological reward affecting, ultimately, the individual’s psychology. Implicit Motives appear as a summary of the whole history of experiences across the early life. [2,4,5,9,15].

**Thesis** **2.**The rewarding effect of any experience should have an effectiveness (intensity) that depends on (i) the individual [4], (ii) the type of motive (achievement, power, affiliation), i.e., physiological system activated [22], and (iii) the moment when the experience occurs (“... the relationship between wanting and liking is an iterative process which after every motivational episode needs to recalibrate, through the outcome evaluation represented by liking, whether less, the same, or more wanting will be appropriate in the future...” [4]).

We propose a mathematical model of the previous ingredients without delving into the physiological aspects of the process, which moreover may help in the parametrization of the model itself. In particular, the model contains parameters which act as proxies for the effectiveness of the reward gained by any new experience.

Due to the multidisciplinary nature of this work, we will use a sufficiently precise presentation albeit avoiding strict mathematical formalism, which can be observed in the references.

### 2.1. The Implicit Motive Space Model

We propose an abstract conceptualization of IM traits as points in a mathematical metric space in order to link the underlying experience dynamics and the individual’s Implicit Motives.

#### 2.1.1. The Elemental Experiences Domain

We define an elemental experience, *e*, as any experience that is solely related to any of the IMs, i.e., Achievement, Power, or Affiliation. Thus, e∈{nPow,nAff,nAch}. For mathematical reasons which will be obvious later, we will identify this set with {1,2,3}, respectively (i.e., *n* Pow = 1 and so on).

One can build sequences of experiences, or *experiential pathways*, $e=(e_{1},e_{2},e_{3},\ldots )$, each $e_{k}$ being an elemental experience from the previously defined domain, (which could be experienced by any individual growing up). Of course, the experiences of any individual are finite in practice. However, we shall see in Section 3 that this fact will not affect the conclusions of the model.

We define the **Elemental Experiences Domain**, *E*, by the set of sequences of elemental experiences. Given two sequences e,f in *E*, the distance
(1)$$d_{E}\left(e,f\right)=\sum_{n=1}^{\infty }\frac{|e_{n}-f_{n}|}{3^{n}},$$
defines a metric in *E* [29].

The metric space $\left(E,d_{E}\right)$ is our mathematical model of **Implicit Motives Experience Space.**

Although the distance just defined might seem rather artificial (in fact, numbers 1,2, and 3 can be re-labeled), it is just a mathematical tool that allows the relationship of two conceptually different (metric) spaces: one corresponding to the experiences and another, the Implicit Motives trait space, related to the psychologic fingerprints introduced in the next section. Theorem 3 will provide the *link* between these two metric spaces.

An element of *E* draws a complete picture of the history of experiences but does not consider the rewarding effect and the respective individual response, which might be decisive in the Implicit Motives generated by a given pathway.

When a sequence of *E* contains only one repeated element, i.e., $e_{k}$ is equal for all *k*, we call these pathways **Absolute Prevailing Experiences** (APE). There are three APE points in *E* (corresponding to the three IM). It would be tempting to relate these three, on one side abstract, points in *E*, with individuals highly biased toward the corresponding implicit motive. However, we can assume that these ideal experiential pathways should yield individuals with prevailing IMs.

#### 2.1.2. The Implicit Motives Trait Space

The individual physiological and psychological responses to experiences should be considered directly related with the corresponding rewards. We shall model these rewarding effects through the **IM-trait space** *X*. We will represent the physiological fingerprints related to the individual experiences as points on *X*. For simplicity, it will be supposed that *X* is embedded in the bidimensional Euclidean space, with the Euclidean distance, *d*; thus, (X,d) is a metric space.

#### 2.1.3. Assumptions

A precise relation between the Implicit Motive Experience Space, *E*, and the IM-trait space, *X*, will be established. To do this, we assume the following:

**Assumption** **A1.**The three Absolute Prevailing Experiences, APE, points in *E* are linked to three distinct points in the IM-trait space, *X*, which we will call **Absolute Prevailing Motives** (APM). For visualization purposes, we can suppose they are the vertex of an equilateral triangle of unit side length (see Figure 1).

**Assumption** **A2.**A rewarding system is represented by a family of random functions $\left\{\varphi_{\theta }:\theta \in \Omega \right\}$ acting on the *IM-trait* space *X*. Any of these functions is, by average, a random contraction and has as a fixed point one of the APM points presented in *Assumption 1*. The contraction ratio, $r_{\theta }$, is related to the intensity of the reward, which is stochastic in the most general version of the model.

**Assumption** **A3.**These functions act in an intermittent manner accruing the individual’s (stochastic) responses to the timeline of experiences, coded by the space *E*. In particular, a probability distribution μ characterizes the frequency and nature of the historical timeline of these experiences.

### 2.2. From the Experience Space to the IM-Trait Space

The idea is that a stochastic process drives the “IM experience–reward dance”. A historic series of experiences generates a sequence of *fingerprints* $\left(x_{n}\right)$ in the IM trait space as $x_{n}=\varphi_{\theta_{n}}\left(x_{n-1}\right)$, using successive draws $\theta_{n}$ from μ. The initial point of the sequence can be chosen randomly, in particular, one of the APM points, as Theorem 1 states.

Here, $\varphi_{\theta }$ represents the response function to the experience θ in the motives set (Achievement, Power, or Affiliation) characterized by its individual-specific (stochastic) reward intensity $\left(r_{\theta }\right)$. In this way, the model’s flexibility is noteworthy.

#### 2.2.1. Implicit Motives as an Attractor

Some might pose a question about the features of IM pathways and whether the sequences of IM-traits in *X*, obtained from the rewards defined by the distribution μ, which codify the event history, draw, in some sense, a long-term configuration structure which may be related to the Implicit Motives.

Theory provides an affirmative answer.

**Theorem** **1.**
*Under the hypothesis above, if the functions $\left\{\varphi_{\theta }:\theta \in \Omega \right\}$ are contractive (“by average”), then there is a unique stationary probability distribution π on X with $P\left(x_{n}\in A\right)\to \pi \left(A\right)$ as n→∞ which does not depend on the starting point $x_{0}\in X$.*


Here, π(A) indicates relative abundance of *fingerprints* in any sub-region *A* of the IM-trait space, defined by the three APM vertex points.

The proof of this theorem is obtained from Diaconis and Freedman [30]. The expression “by average” is stated there in a technically precise manner.

In mathematical terms, the functions in *Assumption 1* should be contractive (statistically). As a consequence, under the frame of this model, Implicit Motives appear as a well-defined structure: the *probability distribution*, π, and its support (the set where the probability measure is concentrated), the *attractor*.

The real essence of the IM construct resides in the attractor which we will call the **IM Entity**.

It is worth mentioning that the existence of the probability distribution π, i.e., the IM Entity, is assured under vast general conditions, in particular the stochasticity of the rewards $r_{\theta }$ and the response functions $\varphi_{\theta }$.

This mathematical tool will provide an interesting set of results, concepts and parameters from complex systems theories, which will become meaningful in the context of psychological Implicit Motives theory, in particular related with the IM stability issue.

#### 2.2.2. A Simplified Working Model

Next, we present a simplified working model in order to introduce entropy-like parameters that will characterize the “IM entity” structure as attractor. It will allow for the visualization of the attractor on the bidimensional Euclidean space.

In the simplified model, the experiences and their effect will be parametrized by means of an **Iterated (Random) Function System** (IFS) [31] and Implicit Motives will appear as attractor of the IFS within the space *X*.

Let $\varphi_{i}:X\to X$ be contractive functions, i.e.:$$d\left(\varphi_{i}\left(x\right),\varphi_{i}\left(y\right)\right)=r_{i}d\left(x,y\right),\;i\in \left\{1,2,3\right\},\;x,y\in X,$$
having as invariant or fixed points the respective APM points, and $r_{i}<1$ standing as the corresponding intensity of the rewarding effect. To provide a more graphic illustration, the reader may assume that $\varphi_{i}$ represents fixed functions instead of random functions and that they are affine maps.

In addition, let $p_{i}$ be probabilities or weights $\left(\sum p_{i}=1\right)$. These discrete probabilities reflect the distribution μ of experiences in the individual’s historic timeline. In practice, these frequencies represent the regime of appearance of the elemental experiences in the individual’s historic timeline.

The set $\left\{\varphi_{1},\varphi_{2},\varphi_{3};p_{1},p_{2},p_{3}\right\}$ is the Iterated Random Function System (IFS), and defines a Markov chain which drives the *experience–reward dance* as follows: (a) take as starting point $x_{0}$, any APM point (or any other point, see Theorem 1); (b) the chain proceeds by choosing, at random, an integer *i* of the index set {1,2,3} with probability $p_{i}$ and moving to $x_{1}=\varphi_{i}\left(x_{0}\right)$. Repeat the random experiment (b). Suppose the new outcome is *j*, then set $x_{2}=\varphi_{j}\left(x_{1}\right)$. By iteration of (b), obtain the sequence $x_{0},x_{1},x_{2},\ldots$ This sequence is called the *orbit* of $x_{0}$.

The algorithm describes a simulation where the functions are chosen randomly according to the probability distribution given by the $p_{i}$ values, which are unique to each individual and correspond to the probabilities of each type of experience of happening. Each motive has an associated $\varphi_{i}$ function, and a probability $p_{i}$ of experiences related to that motive happening. If the motive has a $p_{i}$ of 1/3, then, on average, the function associated to it will act 33% of the time.

Even with this simplified IFS scheme, the approach might be relaxed in several directions. In particular, the functions may be contractive random functions instead of fixed functions. The probabilities $p_{i}$ may be random numbers, and the condition $\sum p_{i}=1$ may be replaced by the average value of the sum $E(\sum p_{i})=1$ (see Hutchinson and Rüschendorf [32]). In addition, the assumption that any experience is solely related to any of the IMs, which may not be plausible, may be circumvented: if an experience is, indeed, related to more than one motive, then the related reward should be obtained by applying all the corresponding rewarding functions instead of just the one.

In all our simulations, the IFS functions take the form:$$\varphi_{i}\left(x\right)=A_{i}x+B_{i},$$
where $A_{i}$ are 2×2 contractive matrices and $B_{i}$ are 2×1 vectors. The elements in $A_{i}$ and $B_{i}$ are chosen so that the function (i) is contractive with ratio $r_{i}$, and (ii) keeps the *i*-th APM point fixed (for simulation purposes, we have added a link to our code at the end of the paper).

The following result provides some geometrical information about the support of the probability distribution π:

**Theorem** **2.***Given the IFS $\left\{\varphi_{1},\varphi_{2},\varphi_{3};p_{1},p_{2},p_{3}\right\}$, there is a unique set K, such that*$$K=\varphi_{1}\left(K\right)\cup \varphi_{2}\left(K\right)\cup \varphi_{3}\left(K\right).$$*This property is called* selfsimilarity.

The proof of this result is provided by Hutchinson [33].

The following result provides the relation between the experiential pathway and the set *K*, and also gives the justification of the term “attractor” of the IFS.

**Theorem** **3.**
*Let $\left(E,d_{E}\right)$ be the IM Experiences Space. For each $e=(e_{1},e_{2},\ldots )$ in E, any x∈X, and any n∈N, let*

$$\phi \left(e,x,n\right)=\varphi_{e_{1}}\circ \varphi_{e_{2}}\circ \cdots \circ \varphi_{e_{n}}\left(x\right).$$

*Then, the limit*

$$\phi \left(e\right)=\lim_{n\to \infty }\phi \left(e,x,n\right),$$

*exists and belongs to K. The function ϕ:E→K from the experiences space to the IM identity attractor thus provided is continuous and onto.*


It is noteworthy that this limit is also independent of the initial point, *x*. This theorem motivated the definition of the distance $d_{E}$ (Equation 1). For measurement purposes, we will use Elton’s ergodic theorem [34]:

**Theorem** **4.**
*If π denotes the corresponding probability distribution provided by Theorem 1 (fitted to the above hypothesis), for any subset B, if $\delta_{B}$ is the indicator function, i.e., $\delta_{B}\left(x\right)=1$ if x∈B and $\delta_{B}\left(x\right)=0$ if x∉B, then*

$$\frac{1}{N}\sum_{k=1}^{N}\delta_{B}\left(x_{k}\right)\to \pi \left(B\right),\;\mathrm{as}N\to \infty .$$



This allows the simulation of the probability distribution on the attractor. If $x_{0},x_{1},x_{2},\ldots ,x_{N}$ is a sequence of points generated by the IFS defined in Section 2.2.2, and $m_{n}$ is the number of $x_{i}$ values that fall inside a given region *B*, then the ratio $m_{n}/N$ approaches the probability that the model assigns to that region, π(B), as the number of iterations *N* grows to infinity. In practice, the estimation of such probability is archived quickly (a number *N* equal to $10^{5}$ suffices).

In particular, we can visualize the answer to the previously stated question (at the beginning of Section 2.2.1) of whether the erratic IM-trait sequence draws a long-term configuration. Figure 1 shows (top left panel) different realizations of the initial iterations for three different IFS, whose parameters are defined in Table 1. Although the picture of these may greatly differ from one realization to another, on the other panels of the figure one can see how the orbit of each IFS stabilizes as the number of iterations grow.

### 2.3. Im Characterization: Uncertainty Entropy-like Parameters

Given a discrete distribution, $p_{i}$, the Information Shannon Entropy, *H*, is calculated as [35]
(2)$$H=-\sum_{i}p_{i}\log_{2}\left(p_{i}\right),$$
provided $p_{i}\log_{2}\left(p_{i}\right)=0$ if $p_{i}=0$.

*H* is expressed in information units (bits) and is a recognized measure of information which has been used in many areas of science with different, but conceptually linked, meanings at diversity, uncertainty, heterogeneity, or entropic level [36].

This context supports the use of entropy. Given that the information would be stored in bits in the brain, it might be wise to expect that brain responses, through spontaneous behavior for instance, should happen according to such storage.

The modeling proposed allows the use of the Information Dimension ($D_{I}$), an entropy-like parameter [37,38], to continuously characterize the distribution that defines the **IM Entity**, and it is related to the heterogeneity or diversity that these profiles may show. This parameter is based in Shannon’s entropy in the following way:(3)$$D_{I}=\frac{-H}{\sum_{i}p_{i}\log_{2}\left(r_{i}\right)}.$$

Please note that $r_{i}$ values must be normalized to calculate this index.

The Information Dimension becomes meaningful in the frame of Implicit Motive modeling: it measures the inherent uncertainty of an unconscious structure which might mediate the individual spontaneous behavior before a new experience (situation). Other meanings, such as diversity of the psychological-mental resources that this model provides, might be possible and have potentially been used in the Implicit Motives setting.

### 2.4. Robustness and Resilience of Implicit Motives

Different sources of changes in Implicit Motives have been discussed [39]. Without going into such discussion, our model also provides some light on this matter through the following result.

**Theorem** **5.**
*Under certain general conditions, for any starting point, the convergence of $P\left(x_{n}\in A\right)\to \pi \left(A\right)$ occurs at an exponential rate. More precisely: For any n, the law of $P_{n}\left(x,\cdot \right)$ is given that $x_{0}=x$ and ρ is the Prokhorov metric used as distance between two probability distributions. Then, $\rho \left[P_{n}\left(x,\cdot \right),\pi \leq B_{x}r^{n}\right]$, and $B_{x}$ and r serve as constants (0<r<1). These bounds apply to all n values and all starting points x.*


A proof can be found in Diaconis and Freedman [30].

Within the frame of this model, this theorem actually explains the resilience property of Implicit Motives. Under the presented hypothesis, if the probability distribution defined by the experiences vector is disturbed, restoration of Implicit Motives occurs at an exponential rate whenever the pattern regime is again re-established.

### 2.5. Illustrated Application with Simulation

As is stated in [9,15], adult motive levels are determined (in part) by how parents treat or influence the early life experiences of their children. For example, the emphasis that parents put on learning bowel and bladder control is significantly related to adult levels of *n* Achievement 25 years later. In addition, allowing very young children freedom in expressing sexual and aggressive impulses is related to adult levels of *n* Power [15].

In this way, while the proposed model parametrizes the individual’s neurophysical response to the different experiences by $r_{i}$, the parameter $p_{i}$ can represent *parenting strategies*, i.e., the relative importance, emphasis, or exposure that parents choose to give to different kinds of experiences which will build, in part, the IM entity; *i* being the motive’s index: i=1 for *n* Pow, 2 for *n* Aff, and 3 for *n* Ach.

A first application of the model that we present here is to illustrate the possible, relative effect of parenting on the individual’s IM identity. We can imagine an hypothetical child, with fixed and equal response to the three types of experiences (i.e., $r_{i}=0.9$ for all motives *i*). This child can be raised using a plethora of parenting strategies, parametrized by different sets of $p_{i}$ values. Table 2 shows the $p_{i}$ values of three different strategies, along with their entropic-like characterization, while Figure 2 (upper and bottom left panels) shows the simulation of the corresponding attractors using $N=10^{5}$ iterations. The first strategy is homogeneous in the sense that no prioritization is given to any type of experience, and the attractor shows a uniform distribution in the support. In this case, the $D_{I}$ parameter attains its maximum possible value, indicating that for this hypothetical child, the highest possible *motivational diversity* among all strategies is reached. The second strategy (upper right panel) gives only 5% of the importance to the first kind of experiences (*n* Pow), while the other two have the same strength. The lower diversity of this strategy, clearly visible in the figure, is corroborated by the $D_{I}$ value of 0.78. The third strategy (bottom left panel) puts 90% of importance on the first motive, yielding a much more heterogeneous IM-identity (bottom left panel) with the lowest $D_{I}$ value of all strategies considered.

One could also ask what is the effect of the neurophysiological response, parametrized by $r_{i}$, on the IM-identity. The bottom right panel of Figure 2 shows the attractor obtained using the previous first parenting strategy with $r_{1}=r_{3}=0.8$ and $r_{2}=0.4$, i.e., the contraction towards the second motive (*n* Aff) has half the value (double the strength) than the other two. This figure also justifies the use of the information dimension, $D_{I}$, instead of the usual entropy, *H*, as the value of *H* would be the same in top left and bottom right panels, while the value of $D_{I}$ in the latter is 0.958.

In summary, any experience leaves a new point on the IM-trait space by means of a function of the IFS (parametrized as above). After a sufficient number of experiences (which mathematically does not need to be that long), a distribution appears on this space. The dynamic of the IFS is limited to the triangle shown in Figure 2, whose vertex is the APM points. When the experiences are mostly related to *n* Pow, the figure will show a great accumulation of points near that vertex, while if the parenting strategy tends to supply experiences of the three kinds equally (strategy 1: $p_{i}=1/3$), then no vertex seems to be favored. The $r_{i}$ parameters (the child’s physiological response to the different kind of experiences) also play a role. We see this in the bottom right graph, in comparison to the upper left. In both, parents use strategy 1, but the child has a lower $r_{2}$ value (associated to *n* Aff) which leads to a bigger accumulation of points in his/her IM map near the *n* Aff vertex. By construction, lower $r_{i}$ values have more effect than bigger ones. This child will have more “mass” of points near the *n* Aff vertex and thus, a stronger *n* Aff motivation level.

## 3. Discussion

According to this model, Implicit Motives emerge as a well-defined robust structure from an apparently erratic sequence of rewards. The probability distribution of IM-traits is supported on the attractor which is ultimately the *Implicit Motive Entity* under the mathematical point of view. The probability distributions theoretically described, which can be easily simulated, are mathematically robust.

The fractal attractor is a type of scale-free structure that appears regularly in self-organized systems. In this case, the hierarchical stratification derived from the model might be consistent with the modular storage of information in the brain [26].

A combination of chance and determinism guides the experience “dance” leading to the creation of such an individual psychological structure. This mimics a somehow alostheric control in the human body that is linked to the ability to adapt and self-manage [40]. The dynamics origin of this stability derived from our model suggests that Implicit Motives may be more accurately described as a homeorhetic state [41].

The attractor and associated probability distribution exhibit *selfsimilarity entropy*: the information content (or else, uncertainty, diversity, heterogeneity) of the measure supported in the rescaled copies of the Implicit Motives support, resembles the one found in the whole attractor (up to the scale). This would mean that relatively short experience pathways might provide similar information (and potentially reliable representation pictures) to the one contained in the whole IM attractor. One wonders if there might exist some relation between this fact and the brain’s neural storage information system.

The fact that finite (or even *short*) experience pathways may provide the whole picture of the IM entity is supported by the fact that orbits have been shown to have the ergodic property [34]. Thus, for practical applications, there is no need for the use of an infinite sequence of experiences.

In particular, the attractor’s fractality would be a structural property resulting from a (simple) method of achieving stability by keeping maximum entropy at any scale [41,42].

Another reason for this intrinsic robustness is that the probability distribution model thrives on randomness. A certain amount of random perturbations are included in the growing process, facilitating the formation of the stable structure rather than hindering it.

The Information Dimension (see Equation (Equation 3)) is an entropy-like information parameter that plays an important role in understanding the heterogeneity features of the homeorhetic state and their interpretation as a kind of *motivational diversity* in the context of the IM construct. The use of this heterogeneity index is justified by the example shown in Section 2.5, as it takes into consideration all the model parameters.

Finally, as indicated in Section 2, Theorem 5 supports the resilience property in a mathematically-based manner. Although this term has been widely used, in this setting it appears as a theoretical result.

Practical applications of this theoretical modeling might lead to the estimations of the individual parameters involved in the equation (i.e., $r_{i}$), possibly by means of physiological research, e.g., a precise design of hormone release. In this sense, it is interesting what was pointed out by McClelland [9] (p. 17):

*“it should be possible to study individual differences in the output of DA [Dopamine] and NE [Norepinephrine] in response to various types of stimulation and to determine just what types of situations are most likely to give rise to increased outputs of one or the other hormone. In this way, we could begin to get an understanding of how motives are formed, based on early affective learning in connection with naturally occurring incentive situations”*.

One possible conjecture, which would be consistent with the IM construct, would be that the individual-specific reward intensities, $r_{i}$, will approach the value 1 as the number of lived experiences grows. That is, the rewarding effect of new experiences diminishes as the learning process advances. It also would be consistent with theories about how the brain learns [27] and with the observation made by McClelland et al. [5] pointing out the parallelism between implicit motive and semantic memory. A reliable picture of the whole attractor, as well as associated parameters, is simulated after a relatively short number of iterations which keep the long-term memory (in other terms, “semantic memory”). Last terms or events (short-term memory), although ignored, do not add much to the whole structure.

Hopefully, the proposed model and the parameters involved might help in the research areas of implicit motives and affective neurosciences [43]. Since neurophysiological methods allow us to measure endocrine and hormone responses related to different motivational scenarios, such measures might serve as indicators of the $r_{i}$ values in the model. Then, even in the case of scenarios of great diversity of experience (the frequencies $p_{i}$ might be supposed to have similar values), entropy-like parameters might provide valuable information on motivational disposition. The model will contribute to understanding individual motivational differences in perceiving these situations and their consequences for biological stress response and subsequent mental health. This will help to design chronic stress measures and treatments adapted to motivational aspects of personality [4,22,44].

Despite the fact that this work aims to be no more than a theoretical contribution, we hope it opens the door to an experimental approach. Following works should address the problem of calibrating the quantitative parameters that the model proposes through IM measurements techniques, both psychological and physiological, to be used in further practical applications (see [4] and references therein).

## 4. Conclusions

The emergence of Implicit Motive as psychological structure is modeled by means of iterated application of individual responses to repeated affective experiences. Under natural translation of key facts of Implicit Motives psychological theory into the mathematical frame, a Random Iteration Function System acting on a metric space is proposed as a dynamical model of experience–reward responses. The model shows how (random) responses produced by intermittent random experiences create a well-defined probability distribution on an attractor which, in this manner, becomes the core of the Implicit Motive entity and a mathematical support of the psychological IM construct. Implicit Motive robustness and resilience properties appear theoretically explained by the model.

The simplified model also provides significant uncertainty entropy-like parameters that may be useful to characterize the psychological structure of an individual IM, an interesting byproduct of the model. Since neurophysiological methods allow the measuring of endocrine and hormone responses related to different motivational scenarios, such measures might serve as inputs in the model [4,18,22]. Thus, the mentioned quantitative tools can potentially be used in empirical works to understand individual motivational differences as well as their consequences [44]. The sum of psychological, physiological, and mathematical facts concerning the issue form a set of resounding arguments supporting the Implicit Motive approach. This mathematical IM model might be helpful to develop new methodologies in this area of research.

## Figures and Tables

**Figure 1 entropy-25-00711-f001:**
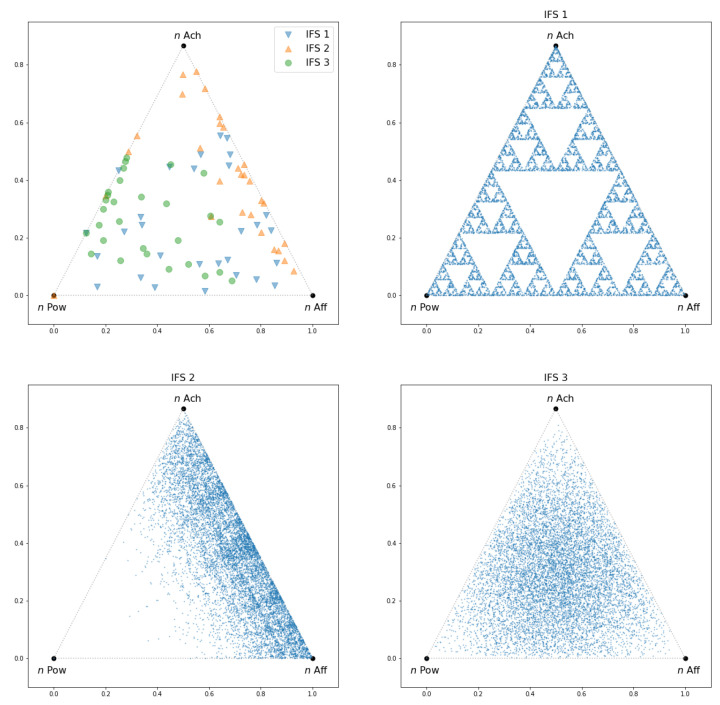
Top left panel shows the 30 first iterations of three different IFS starting from the same APM point (*n* Pow, bottom left edge of the triangle). Notice that no structure is revealed after a few iterations. The other panels show how the orbits of those IFS’s stabilize into a long-term configuration after $N=10^{5}$ iterations. In all cases, the APM points are placed in the vertex of an equilateral triangle of unit size.

**Figure 2 entropy-25-00711-f002:**
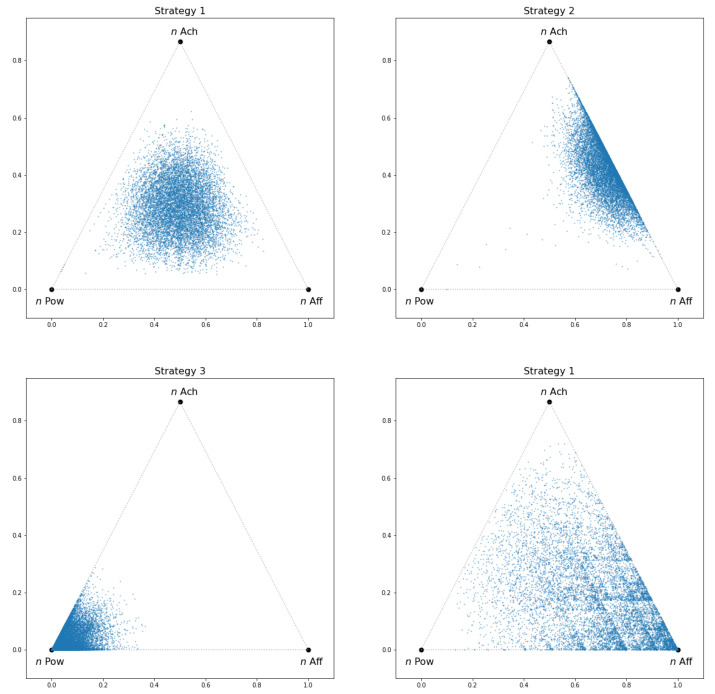
Simulation of the IM attractor for the three parenting strategies described in Table 2. The model proposed generates clearly different IM identities that are affected by the parent strategy. Top and bottom left panels have $r_{i}$ values equal to 0.9. Bottom right panel has $r_{2}=0.4$ and $r_{1}=r_{3}=0.8$, using the same $p_{i}$ values as upper left panel.

**Table 1 entropy-25-00711-t001:** Contraction ratios for the affine functions used in the simulations of Figure 1. The values of $p_{i}$ are equal to 1/3 for all IFSs and all *i*. As an example, for the first IFS, the affine functions use the following matrices: $A_{i}=r_{i}Id_{2\times 2},\;B_{1}=\left(\begin{array}{c} 0\\ 0 \end{array}\right),\;B_{2}=\left(\begin{array}{c} 1/2\\ 0 \end{array}\right),\;B_{3}=\left(\begin{array}{c} 1/4\\ \sqrt{3}/4 \end{array}\right)$.

IFS	$r_{1}$	$r_{2}$	$r_{3}$
1	1/2	1/2	1/2
2	0.9	0.55	0.6
3	3/4	3/4	3/4

**Table 2 entropy-25-00711-t002:** Model parameters and entropy-like characterization of the IM-identity using the Information Dimension, $D_{I}$, of three parenting strategies for a hypothetical child with $r_{i}=0.9$ for all *i*.

Strategy	$p_{1}$	$p_{2}$	$p_{3}$	$D_{I}$
1	0.333	0.333	0.333	1.000
2	0.050	0.475	0.475	0.780
3	0.900	0.050	0.050	0.359

## Data Availability

The figures in this paper were generated using Python on a regular laptop. A jupyter notebook with the code can be found in the following GitHub link (accessed on 1 March 2023).

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
