# Peer review of "Psychological Implicit Motives Construct as an Emergent Fractal Attractor from Intermittent Neurophysiological Responses: Simulation and Entropy-like Characterization"

_entropy, 2023, doi:10.3390/e25050711_

Round 1
Reviewer 1 Report
This is an interesting paper. It provides a suggestive connection between two distant fields, namely the psychology of human motivation and the mathematics of complex systems. It fits very well in the scope of an interdisciplinary journal.
The authors propose a stylized model for an individual’s Implicit Motives construct as a self-similar fractal distribution supported on a triangle. The authors need to assume that Implicit Motives built on repeated affective experiences provide satisfying rewards via neurophysiological responses. This key hypothesis seems justified within the literature on the psychological theory of Implicit Motives. Then a careful interpretation of precise mathematical results within the psychological framework of Human Implicit Motives leads to mapping an individual's Implicit Motives construct as a geometric probability distribution. This seems an intriguing idea for an underlying mechanism giving rise to these psychological structures.
Furthermore, the fractal model naturally provides significant entropy-like parameters that can serve to characterize the psychological structure of implicit motives relative to the input parameters in the model. This is an interesting byproduct of the model, it provides quantitative tools that can potentially be used in empirical work to characterize the motivational system.
The document is clearly written and the style is suitable for a wide audience. Some typos should be corrected in the final edition.
Reviewer 2 Report
In this manuscript the authors provide a mathematical framework that aims to explain the emergence of implicit motives as interindividual differences between individuals, including their prescribed attributes such as their basis in early childhood experiences, their robustness (stability across life) and resilience. I found their approach stimulating and relatively well-grounded in psychological theory. I like that they introduced an entropy-like parameter to characterize the resulting long-term configurations quantitively. I am writing this review from the perspective of a psychologist who conducts research on implicit motives with a special interest in methodological approaches, however my knowledge on the mathematical aspects of the paper is limited. Hence, these aspects should be additionally reviewed in detail by someone who has expert domain knowledge on fractals and iteration random function systems.
In my review I have some questions and suggestions for improvement:
Major
1) The most important point to me is that the manuscript lacks details on the simulations the authors conducted and their interpretation.
a. It took me some time to grasp what exactly is sampled in the 105 iterations done for the illustrative simulation, particularly figure 2. Sure, the text states it’s the “attractors”, but what are these specifically, and what do the numbers on the axes correspond to (on p.3 it is said that for simplicity a bi-dimensional Euclidean space is used, but what are the two dimensions of this space? The psychological and physiological responses?). I assume a part of the truth is also hidden in this sentence on p.4: “Here, p(A) indicates relative abundance of fingerprints in certain sub-region A of the IM-trait space, defined by the three APM vertex points.”? I do not quite understand what this means (is it right to say that the attractor is the parameter A, which is sampled from the probability distribution p, and which denotes a region in the space plotted in the figures? If that is so, this could be made much more explicit). Maybe I am wrong, and this sentence has nothing to do with the question I asked in my first sentence. I think I am also confused because the attractor is called IM Entity which is set equal to the probability distribution p itself on p.5, so it is NOT the parameter A? Whatever the case, the explanation of the attractor / variable(s) sampled need more explanation.
b. Similarly, while I understand the author’s approach on a conceptual level, I do not quite understand what the long-term configurations shown in figure 1 really tell me (besides them being quite beautiful). What is the interpretation of the IFS1 having such a “fractal” shape? I guess it shows a specific configuration of implicit motives in an individual, but what kind of configuration? Is it an accurate interpretation that the configuration of IFS3 is a rather balanced realization of implicit motives in an individual? How do the three IFS look like that led to these configurations? In sum, it would be great if the authors would explain more how these plots emerged and how they can be verbally interpreted.
c. Back to figure 2: Do I understand it correctly that the different parenting strategies are different IFS realizations? Here again, a verbal interpretation would help, for instance when the bottom right panel is described: Does it mean that the configuration is less diverse, because affiliation experiences have a higher reward intensity (by design) hence there are more realizations of x toward the location of the affiliation motive?
d. Please provide details on the technical implementation of the simulation (what software did you use, which packages in which version, etc.).
e. I believe strongly in the value of openness and transparency. In the spirit of open science, I would like to ask the authors to consider sharing a public link to the simulation code, for example as a supplementary file or on the osf (osf.io), to make their analyses better comprehensible and conceptually reproducible. This allows peers to replicate results or investigate details of the simulation (such as experimentally investigate certain parameters as the authors call for in the discussion), thus increasing the value of the study, from which both the academic community as well as the authors benefit. The standards of the Peer Reviewers’ Openness Initiative (https://opennessinitiative.org/) are a helpful starting point to deal with this request if the authors are not sure on how to approach the topic. I am convinced that science must take the issues surrounding replicability of results, availability of materials, scripts and data, and transparency much more seriously going forward, and I am trying to do my part to change scientific standards in that direction, even if some journals have not yet made the decision to require mandatory analysis scripts and data supplements. In this vein, please consider making your code already open at the point of submission in the future, so me and fellow reviewers can access it at this stage.
2) Guidance through the text and mathematics
a. The manuscript would benefit greatly from an overview or an illustration or something similar that helps to guide through the mathematical steps that lead from the implicit motive trait space to the implicit motive (experience) space, and how the different functions and theorems / IFS and introduced key terms play into this.
b. Similarly, more key terms should be printed in bold when they are introduced, such as “Elemental Experience domain” or “Absolute Prevailing Experience” or “Iterated (Random) Function System”.
c. Also, an overview of which variables denoting what would be super helpful (i.e.: m = probability distribution of the reward event history; q = experience with a specific reward intensity; jqn = response function to the experience q; rq = individual specific (neurophysiological) reward intensity; A = attractor (?) = support (?); p = …; W = …; and so on).
d. I think the terms “(metric) space” and “domain” are not used consistently: On p.3 E is introduced as a domain with the corresponding metric space (E,d). However, later, X is called a space as well, whereas accordingly it would be consistent to name X the domain with the corresponding IM-trait (metric) space (X,d). On p.4 (first sentence) E is then called “Space” as well, which should be “domain” according to p.3.
e. How does the combination of the experiences e and the reward intensity rq result in q by the random functions j? I do not understand what a random function is in this context, does it mean it is undefined? Or stochastic in the sense of drawn from a distribution of functions (which would be necessary for the simulation, or not? If so, how does this distribution look like? Is it different for each motive and/or individual)?
f. Why do the authors use “p” instead of “q” on p.5? I got the impression that these variables serve the same purpose. Or is it because “q” denotes a random number whereas “p” is a concrete realized number?
3) The authors use many technical/mathematical terms (also in the title!), which are not at all explained in the main text, such as “fractal” or “orbit” or “contraction ratio” or “support” or “alostheric” or “homeorethic”. Please provide short explanations so that readers from other disciplines can follow.
4) I think the assumption that “any experience is solely related to any of the IM” (p.3) deserves discussion, as this is not plausible: there are many experiences which are relevant for multiple motives at once. I understand that certain assumptions are necessary, but these should be discussed as a limitation.
5) On the distance of the metric space E:
a. Why is it divided by 3^n? The authors state the “we shall see later the usefulness of [the rather artificial or sophisticated definition]” (p.3), but I did not find where it was explained later.
b. If I understand it correctly, as the authors identify the motives with {1,2,3}, the distance between different elemental experiences are scored very differently depending on the motive, as 3-1 (achievement – power) gives a greater number (2) than 2-1 (affiliation – power) = 1. But why should the distance between an achievement and power elemental experience be larger than the distance between an affiliation and power elemental experience?
6) The authors present the study of McClelland & Pilon (1983) as evidence between parental praise and adult nAff, whereas the study only showed this relation for the need for intimacy. As far as I know there is no conclusive evidence that parental practices are related to adult affiliation (see also Heckhausen & Heckhausen, 2018).
7) While I read into another article by the authors which they cite in the current paper (citation 38: Martín, Crawford, Neal, & García-Gutiérrez, 2021, https://doi.org/10.1142/S0218348X21502108), I noticed that the methods of that other article contained verbatim (or nearly verbatim) passages of the paper presented here, for example:
“Some might pose a question about the features of gene pathways and whether the sequences of configurations obtained from initial configurations in the space X draw, in some sense, a long-term configuration structure of the gene “abundant distribution (measure)”. Theory provides an affirmative answer.” (the other article)
Vs.
“Some might pose a question about the features of IM pathways and whether the sequences of IM-traits in X, obtained from the rewards defined by the distribution m, which codify the event history, draw, in some sense, a long-term configuration structure which may be related to the Implicit Motives. Theory provides an affirmative answer.” (the current paper)
Or
“In particular, the functions may be contractive random functions instead of fixed function. The probabilities pi may be random numbers, and the condition Σ pi = 1 may be replaced by the average value of the sum E(Σ pi) = 1.“ (p.9 in the other article, p.6f in the current paper)
There are more similarities like this. While I personally have no strong feelings about the re-use of text an author produced themselves when it is sensible, I would always like to see a transparent cite/mention of the re-usage of text. Further, the guidelines of MDPI include the following: “Manuscripts must be original and should not reuse text from another source without appropriate citation.”
Minor
8) The authors introduce “bold f” as a new sequence on p.3, however only “bold e” was introduced earlier. I would suggest using “bold e1” and “bold e2” to avoid introducing unnecessary new denotations.
9) On p.3 the authors write that “It would be tempting to relate these three, on one hand abstract, points in E, with individuals highly biased towards the corresponding implicit motive. However, we can assume that these ideal experiential pathways should yield individuals with prevailing IMs.” I do not understand why the authors use “However” here, because it describes exactly what the authors did, no? The three APM points in E are related to individuals biased towards these motives (= individuals with prevailing IMs) through ideal experiential pathways?
10) Reading a little bit into contractive functions I got the impression that the “=” (equals) in the first formula on p.5 maybe should be a “£” (less or equal than)? Or is it due to the simplification that is an equal sign?
11) I found the use of the terms “homogenous” and especially “heterogenous” a little bit misleading in the description of the entropic-like characterization of the applied illustration on p.8: Intuitively, I would think a “more heterogenous” IM-identity is an identity that is biased towards a diverse set of motives, while a “more homogenous” IM-identity is biased towards only a few motives. I understand why the authors use “homogenous” to describe a relatively balanced distribution, but I think it can be confusing that in consequence “heterogenous” is an unbalanced distribution towards one motive. Maybe the authors can think of more non-ambiguous terms, perhaps something like “diverse IM-identity” and “specific IM-identity”?
12) The authors discuss on p.10 that their model is consistent with McClelland pointing out the parallelism between implicit motive and semantic memory. Please elaborate on this.
13) Implication of their work on p.10:
a. The authors propose that their model might be useful for stress and support research. Please elaborate how exactly this might be the case.
b. The authors propose that the entropy parameters might be useful in combination with neurophysiological methods. Here also, please elaborate how exactly this might be the case.
14) Missing references: I would like to see citations for:
a. “The Complex System approach” in the first sentence
b. The claim that “Implicit Motives […] have a great influence in human behavior.” on p.1
c. “Motive Disposition Theory” on p.2
d. “Many studies relate each need’s strength in adulthood to child’s rearing practices.” On p.2 (I am not aware of “many” studies that do this).
15) Language and related issues:
a. The authors use abbreviations inconsistently. For example, in the abstract the abbreviation “IM” is introduced, but later in the abstract the abbreviation and the written-out term are used both. Generally, I would not recommend introducing an abbreviation in the abstract. When the authors wish to introduce this abbreviation, I would suggest to only do so in the main text and to use it consistently afterwards (however, I personally would just use the written-out term throughout the text).
b. Some assumptions about implicit motives are presented as facts, where I would use more nuanced writing, for example on p.1: “These motives operate outside of conscious awareness…” could be reworded as “These motives are assumed to operate outside of conscious awareness.” or similar; and “There are three different and fundamental needs …” is a little bit misleading as it makes it sound as if there are only three motive systems overall, whereas achievement, power and affiliation are only the three most focal and well researched ones (however, there is for instance also a sex or a hunger motive, see Schultheiss & Brunstein, 2010, p. xix).
c. The authors’ writing about implicit motives sometimes sounds a little bit off to me.
i. For example, “…the emergence of IM psychological structures” would read more natural as “…the emergence of IM _as_ psychological structures” in the abstract and on p.10
ii. Most uses of “the” before the term may be omitted (e.g., in the Abstract and on p.2: “…characterize the Implicit Motives” -> “…characterize Implicit Motives”; or all on p.2: “… how the IM are built” -> “… how IM are built”; “Once the IM is developed…” -> “Once IM are(?) developed…”; “… leading to the Implicit Motives …” -> “… leading to Implicit Motives …”; “The IM appear…” -> “IM appear…”).
d. I would avoid making the contrast of “exact sciences” versus “other types of sciences” (p.2). Ironically, I think it would be more exact ;-) to talk about “a mathematical point of view”.
e. The Picture Story Exercise is a proper name and should be capitalized (p. 2)
f. On p.3 “on the other hand” is used but without using “on the one hand” somewhere before (which always go together, as far as I know).
g. I think these are typos: “prase”, “aggresive”, and “thesis” (should be plural, no?) on p.2; “historic” on p.4; “his results” on p.5; “it’s” instead of “its” (used two times on l.286 and l.290) on p.8
h. I think these two sentences on p.10 are not grammatically correct: “Practical applications of this theoretical modelling might lead are subscripted [???] to the estimations of the individual parameters involved in the equation (ie. ri), possibly by
means of physiological research, e.g. a precise design of hormone release. In this sense, it is high illustrating interesting [???] what was pointed out by McClelland […]”.
i. It should be explained what “DA” and “NE” are in the citation of McClelland on p.10.
j. Missing words:
i. An “on” is missing in Assumption 2: “Any of these functions is, by average, a random contraction and has as a fixed point [on] one of the APM points presented in Assumption 1.”
ii. An “a” is missing in these two sentences on p.4: “A rewarding system is represented by [a] family of random functions …”; “Here, p(A) indicates relative abundance of fingerprints in [a] certain sub-region A of the IM-trait space, defined by the three APM vertex points.”
iii. I think there should be a “-“ between “entropy” and “like” in the title of section 2.3 (i.e. “entropy-like”).
Reference used in the Review:
Heckhausen, H.; Heckhausen, J. Motivation and development. In Motivation and action; Heckhausen, J.; Heckhausen, H., Eds.; Springer: Cham, Switzerland, 2018; pp. 679–743.
Mcclelland, D.C.; Pilon, D. Sources of adult motives in patterns of parent behavior in early childhood. Journal of Personality and Social Psychology 1983, 44, 564–574. https://doi.org/10.1037/0022-3514.44.3.564.
Schultheiss, O.; Brunstein, J. Implicit motives; Oxford University Press: New York, NY, 2010.
Round 2
Reviewer 2 Report
I thank the authors for their careful attention to my questions, especially given the tight timetable of the journal. Most of my questions were resolved by the author’s responses and revisions, but a few aspects remain. In particular, overall, I appreciate how the authors really clearly explained many aspects that were unclear to me, only to find me somewhat disappointed that they only explained them to me in the response letter, and oftentimes not included them in the paper (see my comments below for where I really would appreciate seeing the responses in the paper as well). I think my job as a reviewer is twofold: a) add domain-level expertise where appropriate, check for errors, quality and transparency b) serve as a prototype reader to highlight any parts that are unclear and can be improved or elaborated on. If these aspects are only explained to me then the paper remains unclear in these aspects for future readers, so I really think that the great explanations from the authors should find their way to the manuscript.
1) Apart from that, my biggest issue is with the author’s justification to not include their code. I looked up the Jupiter notebook they included in the response letter, and I am convinced that it is of value to include in in the paper. Here are some arguments:
a. It helped me understand what the authors did in the simulations, better than solely the verbal and mathematical explanation in the paper. If I am not an absolute exception, then the authors can enhance the understandability and ultimately the impact of their paper when they include their code.
b. The authors argue in the paper that they want to spark interdisciplinary future work from their approach, for example by calibrating the quantitative parameters in experimental work (p.11): It will be more likely for this kind of interdisciplinary collaboration to start, when it is being made easy to “check” back experimental results against simulations, that is, when researchers can run and adapt the code themselves, without first figuring out how to run the simulations. Again, ultimately resulting in more impact (i.e., citations).
c. You argue yourself that the code does not only include the for-loop, but also the code for the visualizations: That’s valuable! Why waste other researcher’s time figuring out how to reproduce nice visualizations, when this time could be spent on experimental research on your models?
d. It is not the job of a research article to provide an exercise for students (at least not its primary job ;-)). Even if one would like to do this, exercises work better if one can compare them with some working solution.
e. You already have the code uploaded. It is not that big of a problem for the comments to be in Spanish (although it would be more accessible in English): Even without the comments the code helps. Hence, it does not produce much work for you, and it certainly doesn’t hurt you or your paper (on the contrary, as I argue in a-b).
2) On my previous comments 1):
a. The authors response on my previous comment 1b) explains figure 2, but not figure 1. Is there a meaningful interpretation for the top right panel in figure 1? What kind of “fingerprint of implicit motives of an individual” does it represent?
b. I absolutely love the authors explanations in the response letter on figure 2: It would be immensely helpful to have this in the paper in section 2.5 ! (“Any experience leaves a new point on the IM-trait space by means of a function of the IFS (parametrized as above). After a long enough amount of experiences (which mathematically does not need to be that long) a figure (distribution) appears on this space. The dynamic of the IFS is limited to the triangle shown in the figure, whose vertex are the APM points. When the experiences are mostly related to nPow, the figure will show a great accumulation of points near that vertex, while if the parenting strategy tends to supply experiences of the three kinds equally (strategy 1: pi=1/3), then no vertex seems to be favoured. Nonetheless, this is not the end of the story, because the ri parameters (the child's physiological response to the different kind of experiences) also play a role. We see this in the bottom right graph, in comparison to the upper left. In both, parents use strategy #1 (equal pi), but the child has a lower r_2 value (associated to nAff) which leads to a bigger accumulation of points in his/her IM map near the nAff vertex. By construction, lower ri values have more effect than bigger ones. This child will have more "mass" of points near the nAff vertex and,thus, a stronger nAff motivation level.“)
3) On my previous comment 2c): Unfortunately, the journal system broke the Greek letters.
a. I will not insist on this but want to reiterate that a table with two columns (the variables in column 1 and a short description in column 2 just as the authors explained it to me in the response letter in their comments on 2c-2f) would be super helpful for all (major) variables introduced. As you could see, I completely misunderstood that A can be any set, and making these things explicit is helpful for readers without a purely mathematical background who still want to follow your technical/mathematical descriptions.
b. In comment 2f) the journal system replaced theta (not psj) with q, so the authors explanation does not really answer my question.
4) Another lovely explanation in the response letter which would be immensely helpful in the paper (maybe in section 2.2) is the following one: “Besides, the functions are chosen randomly according to the probability distribution given by the pis, which are unique to each individual and correspond to the probabilities of each type of experience of happening. Each motive has associated a φ function, and a probability p_i of experiences related to that motive happening. If the motive has a pi of 1/3then, on average, the function associated to it will act 33% of the time.”
5) In the author’s response to my previous comment 3) on the mathematical terms:
a. The authors stated that they added short explanations on “orbit” and “support”: I did not find these explanations in the paper.
b. Also, the fact that something can be looked up on the internet is in itself no argument to refrain from adding it to the paper. I see the tension between readability and making sure that the paper does not get too long / sparsity. However, especially for interdisciplinary work I see great value in guiding the reader through the text instead of confusing them with unknown terms. Again, the authors very short explanations in the response letter are perfect, why not include them in the paper?
6) The author’s explanation in the response letter on my previous comment 4) how to circumvent the assumption that “any is experience is solely related to any of the IM” would fit beautifully to the section where the authors explain how “the approach might be relaxed in several directions.“ (p.5)
7) On the use of heterogenous and homogenous (my previous comment 11): I can follow the reason the authors use these terms, but I think in their explanation to me they actually get confused by these terms themselves, as they state “When the diversity in pareting strategies grows, and the p_i values are different we use the term hetero-genous”. But with Strategy 3 (which results in a “heterogenous” identity” according to the authors) there isn’t diversity in parenting strategies: The parents predominantly use p1 in this strategy, and very seldom p2 or p3: This is not “diverse”! And in consequence, the resulting identity is not “diverse” or “heterogenous”, but has a very specific “mono”-configuration. Hence, maybe it is still worth considering using other terms?
8) The author’s response to my previous comments 12) & 13a) & 13b):
a. I would wish to see more explanations on aspects 12 and 13a: How _exactly_ is their model similar to McClelland’s observation? And what are “the other features of the attractor” besides their robustness that make them similar? How exactly will the model contribute to an improved understanding for consequences in stress responses? These are just empty phrases if not explained / described in detail.
b. I wrote these three comments having in mind that you would elaborate on them in the paper not only in the response latter. These are questions that readers from psychology will very probably ask themselves, so please include elaborations (more than what is currently in the response letter) in the paper.
9) On my previous comment 15b): There is also the implicit motive for intimacy, or partner-related implicit motives of agency (comprising independence) and communion (see e.g., Hagemeyer & Neyer, 2012) so even *within* the IM motive system, there are more than the three discussed motives of affiliation, power, and achievement. Hence, this sentence on page 1 should be revised (it would suffice to add a “major”, e.g. “three major fundamental and difference motives”).
10) The authors forgot to replace the “REFERENCES” placeholder on page 5 with actual references.
Reference used in the Review:
Hagemeyer, B., & Neyer, F. J. (2012). Assessing implicit motivational orientations in couple relationships: The Partner-Related Agency and Communion Test (PACT). Psychological Assessment, 24(1), 114–128. https://doi.org/10.1037/a0024822
Round 3
Reviewer 2 Report
I thank the authors for adding the suggested explanations to the text, and especially for referencing their uploaded code. I have no further comments. Thank you for this interdisciplinary work.